Workshop at the 6th Symposium on Advances in Approximate Bayesian Inference (non-archival), 2024 1–13

# How Useful is Intermittent, Asynchronous Expert Feedback for Bayesian Optimization?

**Agustinus Kristiadi**                             AKRISTIADI@VECTORINSTITUTE.AI
*Vector Institute*

**Felix Strieth-Kalthoff**                          F.STRIETHKALTHOFF@UTORONTO.CA
*University of Toronto*

**Sriram Ganapathi-Subramanian**        SRIRAM.SUBRAMANIAN@VECTORINSTITUTE.AI
*Vector Institute*

**Vincent Fortuin**                           VINCENT.FORTUIN@HELMHOLTZ-MUNICH.DE
*Helmholtz AI, Munich*

**Pascal Poupart**                                      PPOUPART@UWATERLOO.CA
*University of Waterloo and Vector Institute*

**Geoff Pleiss**                                  GEOFF.PLEISS@VECTORINSTITUTE.AI
*University of British Columbia and Vector Institute*

## Abstract

Bayesian optimization (BO) is an integral part of automated scientific discovery—the so-called self-driving lab—where human inputs are ideally minimal or at least non-blocking. However, scientists often have strong intuition, and thus human feedback is still useful. Nevertheless, prior works in enhancing BO with expert feedback, such as by incorporating it in an offline or online but blocking (arrives at each BO iteration) manner, are incompatible with the spirit of self-driving labs. In this work, we study whether a small amount of randomly arriving expert feedback that is being incorporated in a non-blocking manner can improve a BO campaign. To this end, we run an additional, independent computing thread on top of the BO loop to handle the feedback-gathering process. The gathered feedback is used to learn a Bayesian preference model that can readily be incorporated into the BO thread, to steer its exploration-exploitation process. Experiments on toy and chemistry datasets suggest that even just a few intermittent, asynchronous expert feedback can be useful for improving or constraining BO. This can especially be useful for its implication in improving self-driving labs, e.g. making them more data-efficient and less costly.

## 1. Introduction

Accelerating scientific discoveries requires rapid turnover between hypothesis generation and their experimental validation. Particularly in the fields of chemistry, drug discovery, and materials science, this has given rise to the paradigm of *self-driving laboratories* (SDLs), which integrate robotic experimentation with data-driven decision-making algorithms (Tom et al., 2024). Key components of a SDL are (i) a *database* that stores experimental data, (ii) an automated *planner* that recommends the next most informative experiment(s) to be performed, and (iii) a *robotic system* that automates the respective experiment (Figure 1). In SDLs, acceleration (vis-à-vis conventional human-centric discovery loops) stems from two main factors: reducing the number of experiments through efficient automated planners, and minimizing the time-intensive human operations, both in experiments and decision-making.

Nowadays, most SDLs use Bayesian optimization (BO, Garnett, 2023; Močkus, 1975) as the planner. BO works by maintaining a belief about the expensive, black-box function (e.g. the property of a molecule) that one wants to optimize, given the observations about the said function so far. This belief is then used to propose a new point where the function should be evaluated next. Recently, there have been efforts to add humans into a BO loop (Adachi et al., 2024; Huang et al., 2023; Savage et al., 2023; Tiihonen et al., 2022, etc.). Intuitively, human experts have knowledge that can be useful to steer BO's exploration-exploitation. For instance, a human expert might have an intuition about what kinds of molecules are "good" for the problem, or a preference for easy-to-synthesize molecules even if they are at odds with the optimization objective. However, current methods for adding humans back into the loop are at odds with

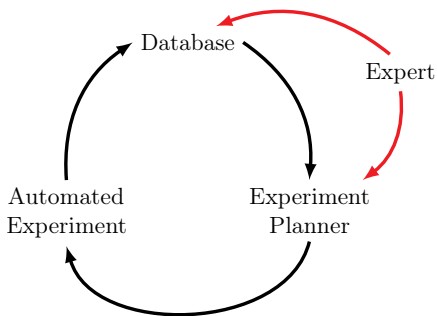

Figure 1: Accelerated scientific discovery takes humans out of the loop (**black**). We investigate the usefulness of human feedback that is incorporated in a compatible way to that spirit (**red**).

the spirit of SDLs. Indeed, prior works incorporate expert feedback either in an offline manner (the BO belief is initialized with a large amount of such feedback) as in Huang et al. (2023) or in an online but blocking manner where the BO loop waits for the expert to input their feedback (Adachi et al., 2024; Savage et al., 2023; Tiihonen et al., 2022).

In this work, we study a setting where incorporating expert feedback is done in an online but non-blocking manner, where the expert feedback is assumed to come randomly. We aim to answer whether *a small amount of intermittent expert feedback throughout the BO loop is effective in steering the exploration-exploitation process*. Crucially, this setting is highly relevant to SDLs since the expert feedback is assumed to be optional—the SDL can still run without any human inputs—and no large offline dataset is required. To this end, we use a Bayesian Bradley-Terry neural network (NN) trained with randomly arriving, sparse expert feedback in an asynchronous manner (i.e. on a different computing thread) w.r.t. the BO loop, in a bid to model human expert preferences. This Bayesian NN is readily available in the BO loop at any given time as a proxy to the true expert preference. The BO loop can then leverage it in its acquisition function to steer its exploration-exploitation process. Our experiments show that even when the feedback randomly arrives 10% of the time, it can still influence BO in meaningful ways. This work thus opens up an avenue for looping humans back into scientific discovery, while preserving the automatic spirit of SDLs.

## 2. Background

### 2.1. Bayesian optimization

Let $f : \mathcal{X} \to \mathcal{Y}$ be a hard-to-compute black-box function. ***Bayesian optimization*** (***BO***; Močkus, 1975) aims to solve the maximization (or minimization) problem $\arg\max_{x \in \mathcal{X}} f(x)$ in a data-efficient manner, by sequentially proposing at which $x \in \mathcal{X}$ should one evaluates $f$ next. The key components of BO are (i) a Bayesian surrogate $p(f \mid \mathcal{D})$ that models our current probabilistic guess about $f$ given the current observations $\mathcal{D} = \{(x_i, f(x_i))\}_{i=1}^m$, and

(ii) an acquisition function (AF) $\alpha : \mathcal{X} \to \mathbb{R}$ that uses $p(f \mid \mathcal{D})$ to "score" each $x \in \mathcal{X}$—the next point to be evaluated is then $\arg\max_{x \in \mathcal{X}} \alpha(x)$. The *de facto* choice for a surrogate $p(f \mid \mathcal{D})$ is Gaussian processes (GPs, Rasmussen and Williams, 2005). Meanwhile, an example of AF is ***Thompson sampling*** (***TS***; Thompson, 1933), which is defined by $\alpha(x) \equiv \hat{f}(x)$ where $\hat{f} \sim p(f \mid \mathcal{D})$. TS has been shown to perform well in applications (González et al., 2017; Hernández-Lobato et al., 2017; Kristiadi et al., 2024)—see also Fig. 7 in Appendix B.

## 2.2. Preference learning

Let $z_0, z_1 \in \mathcal{Z}$ be arbitrary variables from an arbitrary space $\mathcal{Z}$ and $\mathcal{D}_{\text{pref}} := \{(z_{i0}, z_{i1}, \ell_i)\}_{i=1}^m$ be a collection of such pairs, where each $\ell_i \in \{0, 1\}$ indicates which of the two $z_i$'s is preferred. The goal in preference learning (Chu and Ghahramani, 2005) is to train a scoring function $r_\psi(z) : \mathcal{Z} \to \mathbb{R}$ parametrized by $\psi$ under $\mathcal{D}_{\text{pref}}$; such that $z_0$ is preferred over $z_1$ if $r_\psi(z_0) > r_\psi(z_1)$. Often, the ***Bradley-Terry model*** (Bradley and Terry, 1952) is used:

$$p(\ell_i \mid z_{i0}, z_{i1}; \psi) := \text{Cat}(\ell_i \mid \text{softmax}([r_\psi(z_{i0}), r_\psi(z_{i1})])) = \frac{\exp(r_\psi(z_{i\ell_i}))}{\sum_{j=1,2} \exp(r_\psi(z_{ij}))}. \tag{1}$$

That is, each of the logits is given by just one function $r_\psi$, but evaluated at each $z_{ij}$'s.[1]

## 2.3. Laplace approximations

Let $f_\theta : \mathcal{X} \to \mathcal{Y}$ be a neural network (NN) parametrized by $\theta \in \mathbb{R}^d$. The ***Laplace approximation*** (***LA***; Daxberger et al., 2021; Laplace, 1774; MacKay, 1992a) fits a Gaussian $\mathcal{N}(\theta \mid \theta_*, \Sigma(\theta_*))$ centered at a local minimum of the regularized loss function $\mathcal{L}(\theta)$, where the covariance is given by the inverse-Hessian $\Sigma(\theta_*) = (\nabla_\theta^2 \mathcal{L}|_{\theta^*})^{-1}$.

During prediction on a test point $x$, one can opt to linearize $f_\theta(x)$ around $\theta_*$. This implies that $p(f(x) \mid x, \mathcal{D}) \approx \mathcal{N}(f(x) \mid f_{\theta_*}(x), J(x)\Sigma(\theta_*)J(x)^\top)$, where $J(x) := \nabla_\theta f_\theta(x)|_{\theta_*}$ is the Jacobian of the NN's outputs. This is called the ***linearized Laplace approximation*** (***LLA***; Immer et al., 2021; MacKay, 1992b) and has been shown to work well as a surrogate function in sequential decision-making problems (Kristiadi et al., 2023, 2024; Li et al., 2023; Wenger et al., 2023). It can thus be used as a drop-in replacement for the standard GP surrogates. Finally, the LA can also be applied to the Bradley-Terry NN model, resulting in a Bayesian surrogate over human preferences (Yang et al., 2024).

## 3. Experiments

Recall that our main goal is to investigate the usefulness of a small amount of intermittent expert feedback that is gathered in an online but *non-blocking* manner w.r.t. the BO loop. To this end, we use this feedback to learn a Bayesian preference model $p(r \mid \mathcal{D}_{\text{pref}})$. Crucially, we separate the BO loop (Fig. 1, **black arrows**) and the feedback-gathering/preference-learning process (Fig. 1, **red arrows**) into two independent computing threads. See Fig. 2 for a summary. This preference model $p(r \mid \mathcal{D}_{\text{pref}})$ is updated in the 2nd thread as soon as the expert inputs their feedback on some pairs generated from the current $\mathcal{D}_t$ (loaded from the 1st thread). It is then stored and can be accessed by the BO thread at any time.

---

1. Equivalently, it can be written as a Bernoulli likelihood (Rafailov et al., 2023).

---

**Algorithm 1:** Bayesian Optimization

---
**Input:** Initial dataset $\mathcal{D}_1 = \{(x_i, f(x_i))\}_i$,
        surrogate $f$, preference-aware AF $\alpha_r$
**for** $t = 1$ **to** $T$ **do**
    Infer $p(f \mid \mathcal{D}_t)$
    Load $p(r \mid \mathcal{D}_{\text{pref}})$ from the 2nd thread
    $x_{\text{next}} = \arg\max_{x \in \mathcal{X}} \alpha_r(x)$
    Compute $f(x_{\text{next}})$
    $\mathcal{D}_{t+1} = \mathcal{D}_t \cup \{(x_{\text{next}}, f(x_{\text{next}}))\}$
**end**

---

**Algorithm 2:** Preference Learning

---
**Input:** Initial $\mathcal{D}_{\text{pref}} = \{(x_{i0}, x_{i1}, \ell_i)\}_i$, pref.
        surrogate $r$, active learning acqf. $\beta$
**repeat**
    Get the current $\mathcal{D}_t$ from the BO thread
    Present top-$k$ pairs to the expert via $\beta$
    Expert gives labels
    Update $\mathcal{D}_{\text{pref}}$
    Update and store $p(r \mid \mathcal{D}_{\text{pref}})$
**until** terminated

---

Figure 2: The setting is comprised of two asynchronous processes. The first one (**left**) runs the standard BO loop with the preference-aware AF (2)—difference to the standard BO in red. The second one (**right**) gathers optional expert feedback by presenting some pairwise comparisons from $\mathcal{D}_t$ to the expert, and updating $p(r \mid \mathcal{D}_{\text{pref}})$.

To leverage $p(r \mid \mathcal{D}_{\text{pref}})$ in steering the BO's exploration-exploitation process, we define the following AF, based on Thompson sampling:

$$\alpha_r(x; \gamma) := \hat{f}(x) + \gamma \hat{r}(x) \qquad \text{with } \hat{f} \sim p(f \mid \mathcal{D}) \text{ and } \hat{r} \sim p(r \mid \mathcal{D}_{\text{pref}}). \tag{2}$$

Intuitively, the modified BO problem can be seen as a multiobjective problem, maximizing the black-box function $f$ and the latent expert preference $r$, and $\alpha_r$ can be seen as a scalarization of these two objectives (Paria et al., 2020). The hyperparameter $\gamma > 0$ can be interpreted as "how much we want to believe the expert's preference".

### 3.1. Setups

We test our framework on four commonly-used toy test functions, following (Eriksson et al., 2019): (i) 10D Ackley function, (ii) 10D Lévy function, (iii) 10D Rastrigin function, and (iv) 6D Hartmann function. Recall from Fig. 2 that our framework is agnostic to the choice of the surrogates $p(f \mid \mathcal{D})$ and $p(r \mid \mathcal{D}_{\text{pref}})$. We thus apply it to a GP with the Matérn kernel (**GP**) and the Laplace-approximated ReLU MLP with 50 hidden units each (**LA**), representing $p(f \mid \mathcal{D})$. Additionally, we use three real-world drug-discovery datasets: (i) Kinase, (ii) AmpC, and (iii) D4 (Graff et al., 2021), which goals are to minimize their docking scores. The GP kernel used in this setting is the Tanimoto kernel (Griffiths et al., 2023). Both the GP and the NN are trained using Adam under their respective loss functions, while $\gamma$ is set to 1 for simplicity. We use the same NN architecture for $p(r \mid \mathcal{D}_{\text{pref}})$. However, this time, we train it and apply the LA on top of it under the Bradley-Terry likelihood.

To simulate expert feedback, i.e. the label $\ell \in \{0, 1\}$ the expert gives to a pair $(x_0, x_1)$, we create a ground-truth scoring function $s : \mathcal{X} \to \mathbb{R}$ that is hidden from $p(r \mid \mathcal{D}_{\text{pref}})$. Then, we use it to simulate the expert feedback: Given $x_0$ and $x_1$, we set $\ell = 0$ if $s(x_0) > s(x_1)$ and $\ell = 1$ otherwise. Note that, this simulated scoring mechanism is done for convenience—in the real world, the expert can label the pairs based on any criteria.

We simulate a real-world situation where the feedback arrives intermittently as follows. At each BO iteration $t$, with probability $p_{\text{fb}}$, we add three expert feedback $\{(x_{i0}, x_{i1}, \ell_i)\}_{i=1}^3$,

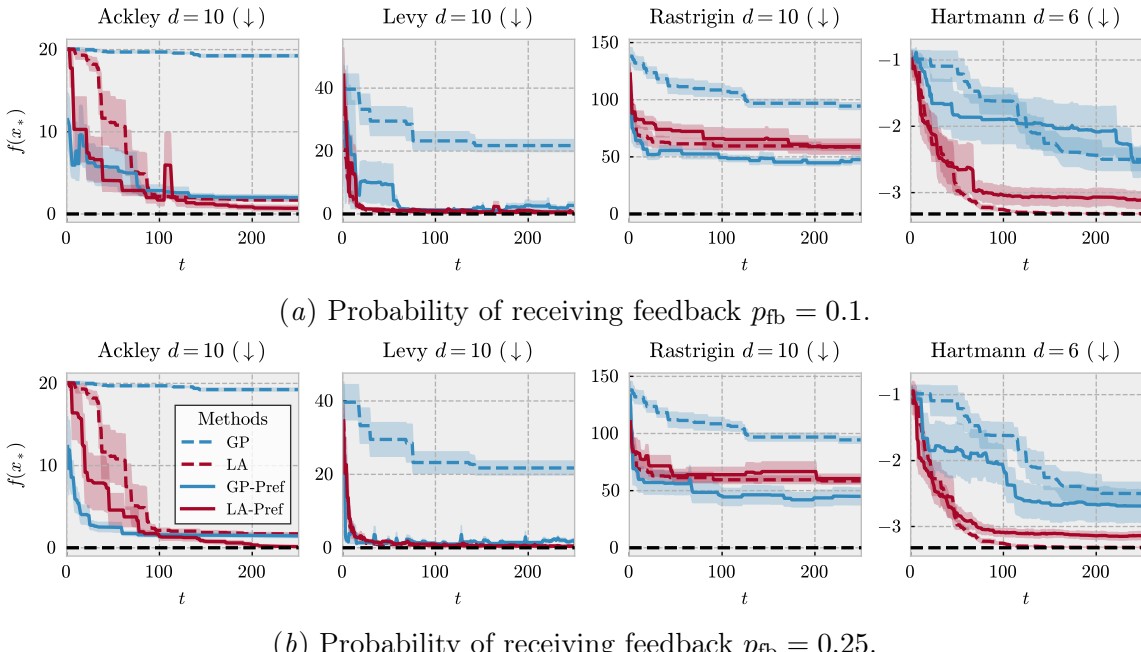

(a) Probability of receiving feedback $p_{\text{fb}} = 0.1$.

(b) Probability of receiving feedback $p_{\text{fb}} = 0.25$.

Figure 3: Results with **random selection** on the expert-feedback thread. Dashed black lines are the optimum values.

where each of the pair $(x_{i0}, x_{i1})$ is proposed by either **random selection** or the **BALD** active learning algorithm (Houlsby et al., 2011).[2]

### 3.2. Results

In Fig. 3, we show the impact of incorporating simulated expert feedback that gives the BO loop some "hints" about the problem at hand through the (hidden) scoring function $s(x) = -\|x - x_*\|_2^2$ where $x_*$ is the ground-truth optimal solution of $f(x)$. We do so to simulate the case where the expert wants to accelerate the BO through their knowledge about the problem. We observe that even when the probability of receiving feedback $p_{\text{fb}}$ is low and no active learning is performed, significant benefits can still be obtained. Specifically, in all test problems, incorporating just a few—with $p_{\text{fb}} = 0.1$, less than 100 randomly-arriving labeled pairs on average—makes the GP surrogate significantly better. We also notice the results are less noisy for higher $p_{\text{fb}}$, i.e. when the expert engages with the expert-feedback thread more often. Similar observations apply to the LA. However, for the LA, in some cases (Hartmann), incorporating expert feedback degrades the BO performances of the LA surrogate. This seems to be caused by the fact that we fix $\gamma = 1$. In the real world, one would want $\gamma$ to be annealed over time; i.e., as more data $\mathcal{D}_t$ is observed, one believes in the experiment results more, instead of their guess about $f(x)$ through $p(r \mid \mathcal{D}_{\text{pref}})$.

In Fig. 4, we present the results under the scenario when the expert feedback is interfering with the BO objective, e.g. when the expert feedback is used to constrain $x$. This is simulated

---

2. Code at https://github.com/wiseodd/bo-async-feedback.

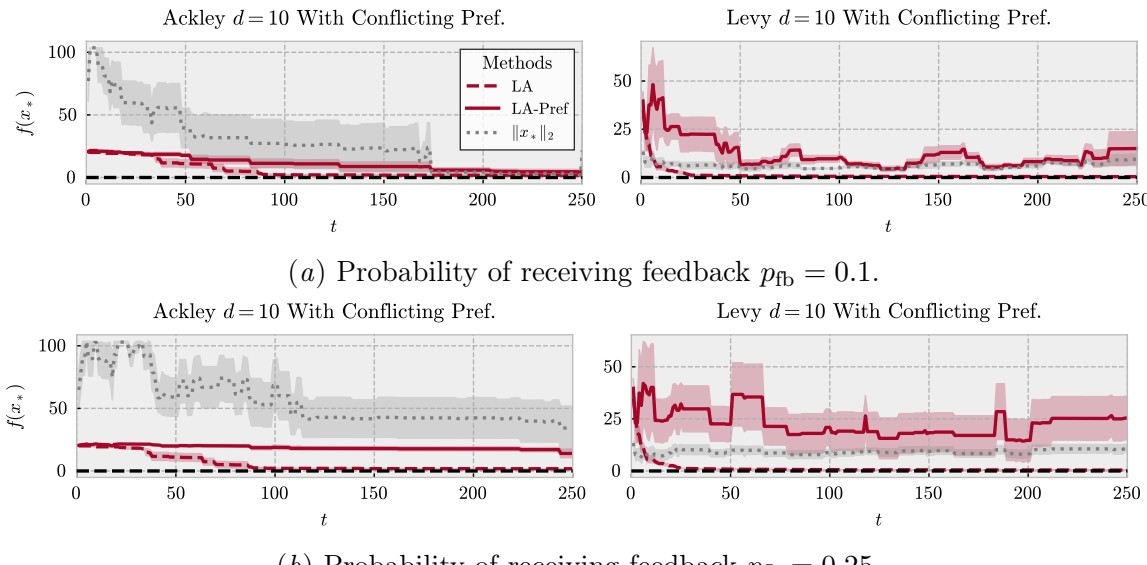

$(a)$ Probability of receiving feedback $p_{\mathrm{fb}} = 0.1$.

$(b)$ Probability of receiving feedback $p_{\mathrm{fb}} = 0.25$.

Figure 4: Toy BO results, simulating expert feedback on $x$ that is conflicting with $f(x)$. The dotted curve represents the norm of the current best $x$.

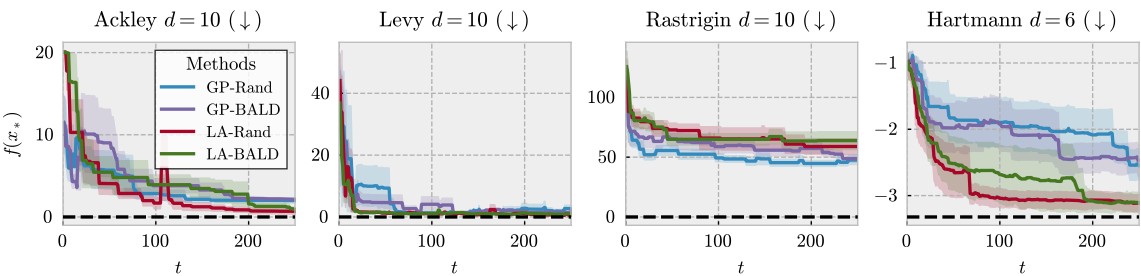

Figure 5: Toy BO results ($p_{\mathrm{fb}} = 0.1$) with **BALD** selection.

by defining $s(x) = -(\|x\|_2 - c)^2$ where $c \geq 0$ is a constant: 100 and 22 for Ackley and Lévy, respectively—note that the norm of the optimal solution is 0 for both problems. We observe the intermittent expert feedback ($p_{\mathrm{fb}} = 0.1$) can indeed steer the BO's exploration-exploitation towards $x$'s with higher norms. We notice that the norm constraint on $x$ is enforced more strongly when more expert feedback is available ($p_{\mathrm{fb}} = 0.25$).

In Fig. 8, we show the effect of incorporating BALD in the expert feedback thread. BALD appears to be not suitable in our setting—its performance is worse than the random selection strategy across the board. Thus, as a future work, it is interesting to develop an active learning algorithm for learning expert preferences specifically, in a bid to reduce the amount of feedback an expert needs to give.

Finally, in Fig. 6, we show results on real-world chemistry (drug discovery) datasets. Here, the expert feeback is simulated by the ground-truth preference model from Choung et al. (2023), which was trained from the preferences of 35 chemists on molecular properties

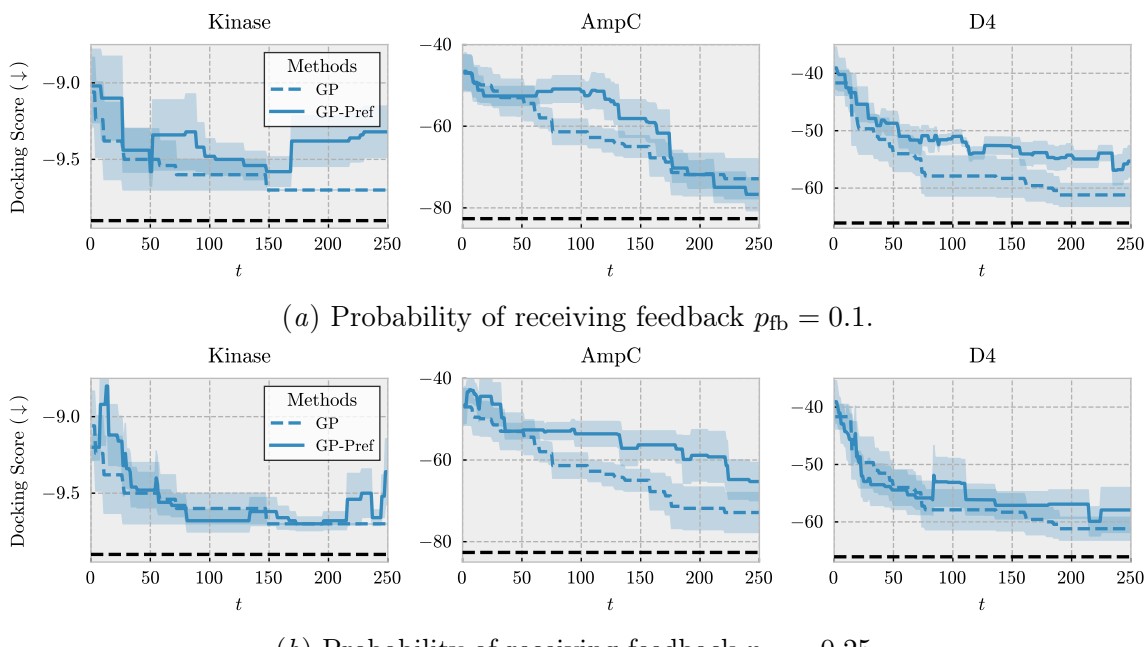

(a) Probability of receiving feedback $p_{\text{fb}} = 0.1$.

(b) Probability of receiving feedback $p_{\text{fb}} = 0.25$.

Figure 6: Real-world chemistry results with **random selection** on the expert-feedback thread. Dashed black lines are the optimum values.

in drug discovery. Our conclusion remains: a small amount of asynchronous, intermittent expert feedback is useful to bias the BO process.

## 4. Conclusion

We have seen that a small amount of intermittent (i.e., randomly arriving over the course of a BO loop) expert feedback that is gathered in an online but non-blocking manner can be useful in guiding a BO's exploration-exploitation. In particular, it can be useful both in accelerating BO's convergence to the optimal solution and to bias BO's solution towards some preferred criteria on $x$. Our results are thus useful for autonomous scientific discovery, to include human experts back into the BO loop, while still preserving the core idea behind self-driving labs. It is exciting to investigate this setting further, especially (i) by allowing the previously-gathered experiment results $\{f(x)\}$ to influence the expert feedback labeling process,[3] (ii) developing a suitable active-learning strategy, and (iii) evaluating it in real-world material- and drug-discovery settings.

---

3. This is a generalization to the current formulation. Useful to counteract the biased belief that an expert might have. E.g. the expert might initially believe that $x_0$ is worse than $x_1$ (i.e. $\ell = 0$), but knowing the experiment results $f(x_0)$ and $f(x_1)$, they might want to prefer $x_1$ instead of $x_0$ (i.e. $\ell = 1$)

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

## Appendix A. Related Work

Human preferences have recently been incorporated on top of standard BO. Huang et al. (2023) proposed a two-stage process where preference learning over pairs of $x$'s is performed and then the learned weights are shared with a BO surrogate. Similarly, Adachi et al. (2024) used a GP-based preference model as an additional prior for the BO process, incorporated via an upper confidence bound acquisition function. However, the two works above assume a relatively large amount of preference data available at initialization. Moreover, Adachi et al. (2024) assumed that the human expert provided their inputs at each BO iteration.

Tiihonen et al. (2022) designed a BO system that can query a human expert when a proposed input $x_{\text{next}}$ is dissimilar to the previously gathered inputs, in a bid to improve the trustworthiness of $x_{\text{next}}$. Meanwhile, instead of using the Bradley-Terry model, Savage et al. (2023) involves humans in the BO loop via a batched BO paradigm, where the expert needs to pick between all the proposed points in the batch. In contrast to those works, the setting discussed in this work leverages preference learning in an online manner, asynchronous to the BO process, and the expert feedback is assumed to be sparse and intermittent.

Human feedback has been asynchronously incorporated into online algorithms. Balsells et al. (2023) and Villasevil et al. (2023) asynchronously incorporate non-expert human feedbacks into reinforcement learning exploration. These feedbacks are being used to learn a distance function from the current agent's state to the goal state. To the best of our knowledge, asynchronous human feedback has not been used to model vague expert intuition, in a bid to steer exploration-exploitation in BO.

## Appendix B. Additional results

### B.1. On Thompson sampling

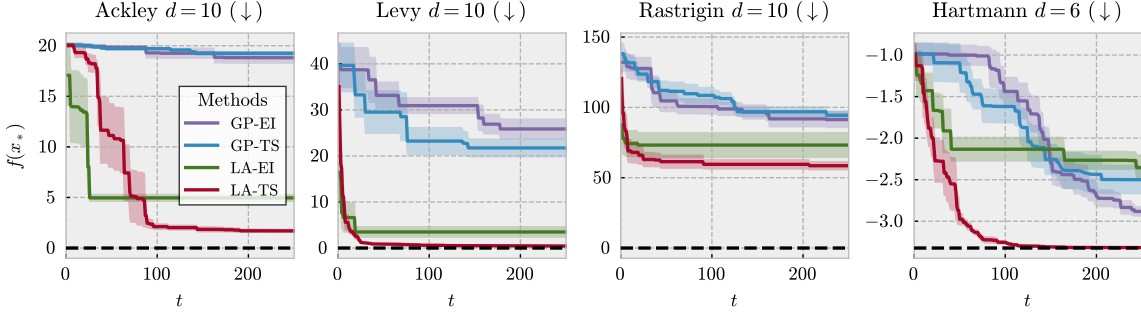

Figure 7:   Thompson sampling (TS) versus expected improvement (EI).

In Fig. 7 we justify our choice of incorporating the preference model through the Thompson-sampling-style acquisition function (2). We found that Thompson sampling works very well for the LA surrogate, indicating that for high-dimensional problems, the LA posterior is more accurate and calibrated than GP's. Meanwhile, we observed a mixed-bag performance of Thompson sampling with GP surrogates.

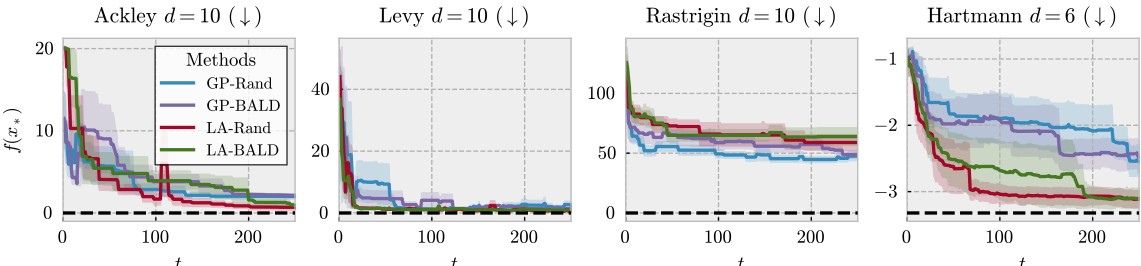

Figure 8: Toy BO results ($p_{\mathrm{fb}} = 0.1$) with **BALD** selection.

## B.2. Active Learning

We present additional results with different active learning acquisition functions to sample the expert feedback. In Fig. 8, we show the effect of incorporating BALD in the expert feedback thread. BALD appears to be not suitable in our setting—its performance is worse than the random selection strategy across the board. Thus, as a future work, it is interesting to develop an active learning algorithm for learning expert preferences specifically, in a bid to reduce the amount of feedback an expert needs to give.

In Fig. 9, we show the effect of picking the top-$k$ pairs where the absolute differences of the Thompson samples $\tilde{r}(x_0)$ and $\tilde{r}(x_1)$ are small. This corresponds to picking pairs with high entropy under the Bradley-Terry model, similar to commonly used uncertainty-reduction active learning strategies. However, we find that the performance is even worse than the random-selection strategy. This further reinforces our findings about BALD in Fig. 8.

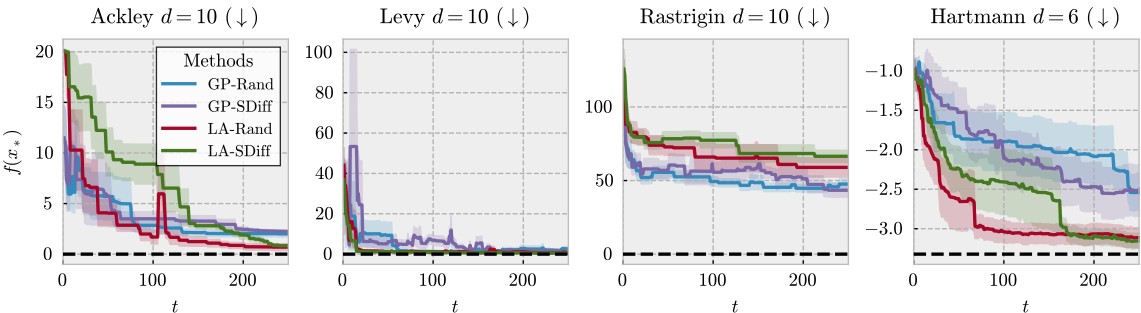

Figure 9: Toy BO results ($p_{\mathrm{fb}} = 0.1$) with an active learning acquisition function where the difference between the Thompson samples $\tilde{r}(x_0)$ and $\tilde{r}(x_1)$ is **minimized**. "SDiff" stands for "small difference".

We also test the opposite in Fig. 10: We pick the top-$k$ pairs where the absolute differences between the Thompson samples are large. While this is at odds with the standard active learning wisdom, we find that the performance is better than the uncertainty-maximizing strategies. However, it is still not significantly better than the random strategy. This highlights the need to develop an active learning strategy that is more compatible with preference modeling both in general and in our setting, where the data are intermittent.

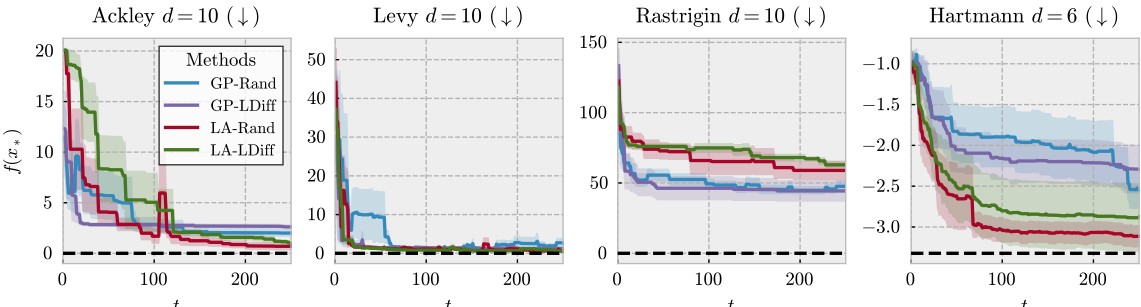

Figure 10: Toy BO results ($p_{\text{fb}} = 0.1$) with an active learning acquisition function where the difference between the Thompson samples $\tilde{r}(x_0)$ and $\tilde{r}(x_1)$ is **maximized**. "LDiff" stands for "large difference".

