# OpenReview forum: "How Useful is Intermittent, Asynchronous Expert Feedback for Bayesian Optimization?"
_approximateinference.org/AABI/2024/Symposium — AABI 2024_

### Official Review · Reviewer_wSDW · 2024-04-13
**Review: How Useful is Intermittent, Asynchronous Expert Feedback for Bayesian Optimization?**

**Rating:** 6
**Confidence:** 4

**Review:**

This paper is about bayesian optimisation centred around the question: if a small amount of expert feedback injected randomly during optimisation will it improve BO performance? I think this idea is interesting and to my knowledge it has not been analysed before in the manner that the authors do. I think the paper is well-written, somewhat thorough and I appreciate the amount of synthetic numerical experiments, however, I do have some concerns:

- It is not clear to me to which extend the results of this paper tells anything about if real-world practical scenarios would benefit from these findings and to which extend. The authors only show synthetic and toy problems, and I think this is the biggest weakness of the paper. Especially since they are arguing for human expert feedback without having any experiment with a human expert giving the feedback. We don't know from these findings if this approach works unless the authors can argue that it is reasonable to assume the human would act in the same manner as their simulated agent, which they have not done.
- I am not sure I get why the expert feedback guides the exploration-exploitation in BO, specifically. The authors did not explain this specific aspect properly. If it only came down to this, could better initialisation or calibrated uncertainties do the same?
- I am slightly surprised that the authors obtain so bad results for a standard GP. I have played around with the same problems and do not remember it not converging that bad - did the authors do hyperparameter tuning for it to be a fair comparison?
- The authors might consider plotting the y-axis on the figures (with offset for negative values) in log scale in order to be able to spot the differences even better.

I think the paper is good enough for this workshop but only marginally. I am willing to increase my score if the authors can argue against my biggest concerns.

---

### Official Review · Reviewer_ejSB · 2024-04-16
**A variation on existing methods**

**Rating:** 5
**Confidence:** 4

**Review:**

The paper considers a learning problem where the objective function is costly to compute, e.g., choosing a point x and carrying out an experiment to find f(x).  The problem arises in materials design, for example, where each new f(x) might correspond to a physical experiment and f(x) measures some desired properties of the material, and the goal is to navigate to an optimal x* with as few experimental samples as possible.  The authors consider the case when a human expert may be exploited, such as for the initial choice of x, or during the experimentation cycle.  They set up a dual cost function in equation (2), that adds a weighted human preference term, where the preference learning is driven by an expert.


Overall, the idea is straightforward.  In many such problems, the evaluation of f(x) might be very slow, e.g., a materials experiment.  In such cases the ‘blocking’ method of human input (or not) makes sense in general.  And, it isn’t clear why the existing methods, such as blocking, can’t be applied intermittently.


The examples are limited to numerical exploration and the simulation of an ‘expert’, and the results don’t seem surprising.  An optimal expert will improve the learning, and a biased expert will bias the learning.  An experiment with a human would be much more interesting.

---

### Official Review · Reviewer_NxqT · 2024-04-21
**Novel approach leveraging intermittent expert feedback to guide Bayesian optimization for autonomous labs, with promising initial results but needs further validation.**

**Rating:** 6
**Confidence:** 3

**Review:**

Quality:
Pros:
- The idea of incorporating intermittent, asynchronous expert feedback into Bayesian optimization is novel and has practical relevance for autonomous scientific discovery.
- The methodology is clearly explained, with good background on relevant topics like Bayesian optimization, preference learning, and Laplace approximations.
- The experimental setup is reasonable, testing on standard benchmark functions and comparing different surrogate models (GPs and neural networks with Laplace approximation).
- The results demonstrate that even a small amount of expert feedback can positively impact BO's performance, either accelerating convergence or biasing solutions based on expert preferences.

Cons:
- The simulated expert feedback may not fully capture real human behavior and intuition.
- There are no experiments on real-world problems from scientific domains like materials/drug discovery.
- The approach to incorporate expert preferences via a simple linear scalarization may be too simplistic.

Clarity:
Pros:
- The paper is well-structured with clear sections for background, methodology, experiments, and conclusions.
- Technical concepts are explained concisely with good use of mathematical notation.
- The motivations, hypotheses, and findings are clearly articulated.

Cons:
- Some details of the experimental setups could be expanded upon for better reproducibility.
- A few derivations and explanations could benefit from some additional clarity (e.g. the Laplace approximation section).

Originality:
The key idea of leveraging intermittent, asynchronous expert feedback to guide Bayesian optimization appears to be a novel contribution. Prior works have mostly focused on incorporating expert preferences in an offline manner or by blocking the BO loop to await synchronous feedback. The proposed asynchronous setup is more compatible with autonomous labs.

Significance:
The work tackles an important problem of including human expertise and intuition into autonomous scientific discovery processes like Bayesian optimization, while still preserving their self-driven nature. If successful, it could lead to more data-efficient and cost-effective optimization for applications like materials/drug design.

However, the significance is limited by the use of simulated feedback functions and lack of real-world evaluations. Demonstrating success on actual scientific problems would greatly increase the significance.

---

### Official Review · Reviewer_yVCp · 2024-04-23
**Interesting work for incorporating human feedback in the BO loop**

**Rating:** 7
**Confidence:** 3

**Review:**

This paper considers a problem of incorporating human expert feedback in the Bayesian optimization loop. Specifically, it tries to answer how much a small amount of feedback can influence the effectiveness of the exploration-exploitation process by introducing the process of preference learning and a different form of acquisition function.

Strength
- The experimental results show the proposed method accelerates the convergence of the Bayesian optimization process
- Incorporating an active learning step seems to be smooth

Weakness
- Curious about how this approach would work for real-world examples beyond toy problems
- Lack of comparison with other baseline works

---

### Official Review · Reviewer_9YV5 · 2024-04-24
**Can asynchronous expert feedback improve Bayesian optimization? (maybe)**

**Rating:** 6
**Confidence:** 2

**Review:**

This submission considers self-driving laboratories (SDLs), wherein an automated system designs and conducts experiments, collects results, and then iteratively uses said results to design the next round of experiments. While this process may be completely autonomous, it's recognized that expert (human) input may help guide this process. This submission addresses the question of how to accomplish this in an asynchronous manner such that the loop can continue online and will not wait for human input. This is accomplished by taking essentially a weighted (by hyperparameter $\gamma$) combination of the current learned preferences and the expert input.

This approach is applied to 4 different test functions. The probability of receiving feedback per iteration is assumed to be either 0.1 or 0.25, and the hyperparameter is assumed to be 1. In general it is shown that providing sparse intermittent feedback is beneficial, though that is not always true as in the case of the Laplace approximation of the 6D Hartmann function (figure 3). It's demonstrated that intermittent feedback can be used to drive the result to prespecified "solutions" (figure 4). It is also shown that the BALD selection approach yields worse performance across the board (figure 5).

Overall, the submission is clearly written; however, the "Related Work" section is relegated to an appendix. While there are clear space constraints in a submission of this type, it may be preferable to include this in the main body to appeal to a wider audience. Similarly, it's mentioned that fixing $\gamma=1$ may be the cause of poor results for the Laplace approximation of the 6D Hartmann, but no further explanation is provided.

Although the model and results are intriguing, it's difficult for this reviewer to see future avenues for this particular work beyond additional simulation studies. In fairness that may reflect more on the reviewer than the submission, and the appeal of SDLs, and the ability to incorporate outside information, is apparent.

---

### Official Review · Reviewer_2ojA · 2024-04-25

**Rating:** 8
**Confidence:** 2

**Review:**

In this paper, the authors studied self-driving laboratories in Bayesian optimization and investigated whether a small amount of randomly arriving expert feedback can improve a Bayesian optimization campaign. They found that adding additional asynchronous expert feedback can help to improve the results. The authors claim that their findings imply a broad range of future work.

In general this paper is well motivated, and clearly structured. To my best knowledge this paper is novel in their line of work. My main question is that I wonder if they can elaborate more on the potential applications they mentioned in the paper.

---

### Meta-Review · Area_Chair_QKaH · 2024-05-27

**Recommendation:** Accept (Poster)
**Confidence:** 4

**Metareview:**

The authors study the role of asynchronous expert feedback in Bayesian optimization. The premise is that for automated scientific discovery, human inputs and feedback should be minimal despite its usefulness. The authors study whether a small amount of feedback arriving asynchronously can help in learning preferences that better steer the exploration/exploitation process. The reviewers felt the paper was well written and the experiments showed promise in understanding means of accelerating convergence in BO. I would have liked to see more motivation for why human feedback should be minimal as opposed to something like a human-AI collaborative system or a mixture of experts type approach where human expertise and model expertise is leveraged for bayesian optimization. I would have also liked to understand what places having human feedback is helpful and if this would remain so in alternative scenarios such as in noisy environments. I would have also liked to have seen more discussion on the assumptions underlying the expert ie are they considered optimal in certain regions and suboptimal in others, what happens if they introduce bias in the process etc. Nonetheless I think the paper has promise so I vote for borderline acceptance.

---

### Decision · Program_Chairs · 2024-05-27

Accept